# Domain-Invariant Projection Learning for Zero-Shot Recognition

**An Zhao**[1,•]  **Mingyu Ding**[1,•]  **Jiechao Guan**[1,•]  **Zhiwu Lu**[1,*]  **Tao Xiang**[2,3]  **Ji-Rong Wen**[1]

[1]Beijing Key Laboratory of Big Data Management and Analysis Methods
School of Information, Renmin University of China, Beijing 100872, China
[2]School of EECS, Queen Mary University of London, London E1 4NS, U.K.
[3]Samsung AI Centre, Cambridge, U.K.

`zhiwu.lu@gmail.com`    `t.xiang@qmul.ac.uk`
•Equal contribution    * Corresponding author

## Abstract

Zero-shot learning (ZSL) aims to recognize unseen object classes without any training samples, which can be regarded as a form of transfer learning from seen classes to unseen ones. This is made possible by learning a projection between a feature space and a semantic space (e.g. attribute space). Key to ZSL is thus to learn a projection function that is robust against the often large domain gap between the seen and unseen classes. In this paper, we propose a novel ZSL model termed domain-invariant projection learning (DIPL). Our model has two novel components: (1) A domain-invariant feature self-reconstruction task is introduced to the seen/unseen class data, resulting in a simple linear formulation that casts ZSL into a min-min optimization problem. Solving the problem is non-trivial, and a novel iterative algorithm is formulated as the solver, with rigorous theoretic algorithm analysis provided. (2) To further align the two domains via the learned projection, shared semantic structure among seen and unseen classes is explored via forming superclasses in the semantic space. Extensive experiments show that our model outperforms the state-of-the-art alternatives by significant margins.

## 1   Introduction

The recent focus on object recognition has been on large-scale recognition problems such as the ImageNet ILSVRC challenge [47]. Since the latest deep neural network (DNN) based models [49, 53, 12, 19] are reported to achieve super-human performance on the ILSVRC 1K recognition task, a question arises: are we close to solving the large-scale recognition problem? The answer clearly relies on how large the scale is: 1) There are approximately 8.7 million animal species on earth; in that context, the ILSVRC 1K recognition task is nowhere near large-scale; 2) Most existing object recognition models (particularly those DNN based ones) require hundreds of image samples to be collected from each object class, but many of the object classes are rare and it is impossible to collect sufficient training samples for some of the rare classes even with the help from social media platforms (e.g., most of the beetle species have never been photoed by amateurs). Therefore, there is still a long way to go before a computer vision model can recognize *all* object categories.

One approach to overcoming the above challenge is zero-shot learning (ZSL) [48, 25, 46, 50, 8, 69, 10, 1, 61]. ZSL aims to recognize a new/unseen class without any training samples from the class. All existing ZSL models assume that each class name is embedded in a semantic space, such as attribute space [22, 25] or word vector space [14, 54]. Given a set of seen class samples, the visual features are first extracted, typically using a DNN pretrained on ImageNet. With the visual feature representation of the images and the semantic representation of the class names, the next task is to learn a joint embedding space using the seen class data. In such a space, both feature and semantic

representations are projected so that they can be directly compared. Once the projection functions are learned, they are applied to the unseen test images and unseen class names, and the final recognition is conducted by simple search of the nearest neighbour class name for each test image.

One of the biggest challenges in ZSL is the domain gap between the seen and unseen classes. As mentioned above, the projection functions learned from the seen classes with labelled data are applied to the unseen class data in ZSL. However, the unseen classes are often visually very different from the seen ones. Therefore, the domain gap between the seen and unseen class domains can be large. Consequently, the same projection function may not be able to project an unseen class image to be close to its corresponding class name in the joint embedding space for correct recognition. To tackle the projection domain shift [15, 23, 45] caused by the domain gap, a number of ZSL models resort to transductive learning [69, 18, 64, 26, 58, 65] in order to narrow the domain gap using the unlabelled unseen class samples. However, without any labels, the unseen class data has limited effect in overcoming the domain gap using existing transductive ZSL models.

In this paper, we propose a novel ZSL model termed domain-invariant projection learning (DIPL). Our model is based on transductive learning but differs significantly from existing models in two aspects. First, we introduce a domain-invariant task, namely visual feature self reconstruction. Specifically, after projecting a feature vector representing the object visual appearance into a semantic embedding space, it should be able to be projected back in the reverse direction to reconstruct the original feature vector (see explanation in Sec. 3.2). By imposing such forward and reverse projection learning on the seen/useen class data, our DIPL model takes a simple linear formulation that casts ZSL into a min-min optimization problem. Solving the problem is non-trivial. A novel iterative algorithm is thus developed as the solver, followed by rigorous theoretic algorithm analysis. Note that the proposed algorithm could potentially be used for solving other vision problems with min-min optimization involved. Second, we align the two domains by exploiting shared superclasses. The idea is simple: although the seen and unseen classes are different, they site in an object taxonomy where the root node is 'object'. Tracing towards the root, the classes in the two domains will share the same ancestors or superclasses. In this work, we take a data driven approach without the need for manually defined taxonomy. Concretely, the superclasses are generated automatically by k-means clustering in the semantic space, which then act as a bridge to align the two domains using our DIPL model.

Our contributions are: (1) A novel transductive ZSL model is proposed which aligns the seen and unsee class domains using domain-invariant feature self-reconstruction and superclasses shared across domain alignment. (2) We formulate ZSL as a min-min optimization problem with a simple linear formulation that can be solved by a novel iterative algorithm. Note that the proposed algorithm could potentially be used for solving other vision problems with min-min optimization involved. (3) We provide rigorous theoretic analysis for the proposed algorithm. Extensive experiments show that the proposed model yields state-of-the-art results. The improvements over alternative ZSL models are especially significant under the more challenging pure and generalized ZSL settings.

## 2 Related Work

**Semantic Space**. Various semantic spaces are used as representations of class names for ZSL. The attribute space [67, 61] is the most widely used. However, for large-scale problems, annotating attributes for each class becomes very difficult. Recently, semantic word vector space has begun to be popular especially in large-scale problems [14], since no manually defined ontology is required and any class name can be represented as a word vector for free. In addition, in [2], the manually-defined object taxonomy was also used to form the semantic space for ZSL. In this paper, although we also leverage superclasses in ZSL, we take a data driven approach based on k-means clustering without the need for manually-defined taxonomy.

**Projection Learning**. Relying on how the projection function is established, existing ZSL models can be organized into three groups: (1) The first group learns a projection function from a visual feature space to a semantic space (i.e. in a forward projection direction) by employing conventional regression/ranking models [25, 2] or deep neural network regression/ranking models [54, 14, 44, 4]. (2) The second group chooses the reverse projection direction [50, 23, 51, 66], i.e. from the semantic space to the feature space, to alleviate the hubness problem suffered by nearest neighbour search in a high dimensional space [42]. (3) The third group learns an intermediate space as the embedding space, where both the feature space and the semantic space are projected to [31, 68, 8]. As a combination of the first and second groups, our DIPL model integrates both forward and reverse projections for ZSL.

More importantly, different from existing projection learning models, our model is also formulated for transductive learning and ZSL with superclasses to address the domain gap problem. Note that *our transductive formulation is non-trivial*, and a novel iterative algorithm is formulated as the solver, with rigorous theoretic algorithm analysis provided.

**Transductive ZSL**. Transductive ZSL is proposed to tackle the projection domain shift [15, 23, 45] caused by the domain gap, through learning with not only the training set of labelled seen class data but also the test set of unlabelled unseen class data. According to whether the predicted labels of the test images are iteratively used for model learning, existing transductive ZSL models fall into two categories: (1) The first category [15, 17, 26, 45, 64] first constructs a graph in the semantic space and then transfers to the test set by label propagation. A variant is the structured prediction model [69] which employs a Gaussian parametrization of the unseen class domain label predictions. (2) The second category [18, 23, 27, 51, 58, 65] involves using the predicted labels of the unseen class data in an iterative model update/adaptation process as in self-training [62, 63]. Our DIPL model can be considered as *a combination of these two categories* of transductive ZSL models.

**ZSL with Superclasses**. There is little attention on ZSL with superclasses. Two exceptions are: 1) [20] learns the relation between attributes and superclasses for semantic embedding; 2) [39] uses the taxonomy to define the semantic representation of each object class. Note that these two methods have a limitation that manually defined taxonomy must be provided at advance. In this paper, our method is more flexible by generating the superclasses with k-means clustering.

## 3  Methodology

### 3.1  Problem Definition

Let $\mathcal{S} = \{s_1, ..., s_p\}$ denote a set of seen classes and $\mathcal{U} = \{u_1, ..., u_q\}$ denote a set of unseen classes, where $p$ and $q$ are the total numbers of seen and unseen classes, respectively. These two sets of classes are disjoint, i.e. $\mathcal{S} \cap \mathcal{U} = \phi$. Similarly, $\mathbf{Y}_s = [\mathbf{y}_1^{(s)}, ..., \mathbf{y}_p^{(s)}] \in \mathbb{R}^{k \times p}$ and $\mathbf{Y}_u = [\mathbf{y}_1^{(u)}, ..., \mathbf{y}_q^{(u)}] \in \mathbb{R}^{k \times q}$ denote the corresponding seen and unseen class semantic representations (e.g. $k$-dimensional attribute vector). We are given a set of labelled training images $\mathcal{D}_s = \{(\mathbf{x}_i^{(s)}, l_i^{(s)}, \mathbf{y}_{l_i^{(s)}}^{(s)}) : i = 1, ..., N_s\}$, where $\mathbf{x}_i^{(s)} \in \mathbb{R}^{d \times 1}$ is the $d$-dimensional visual feature vector of the $i$-th image in the training set, $l_i^{(s)} \in \{1, ..., p\}$ is the label of $\mathbf{x}_i^{(s)}$ according to $\mathcal{S}$, $\mathbf{y}_{l_i^{(s)}}^{(s)}$ is the semantic representation of $\mathbf{x}_i^{(s)}$, and $N_s$ denotes the total number of labelled images. Let $\mathcal{D}_u = \{(\mathbf{x}_i^{(u)}, l_i^{(u)}, \mathbf{y}_{l_i^{(u)}}^{(u)}) : i = 1, ..., N_u\}$ denote a set of unlabelled test images, where $\mathbf{x}_i^{(u)} \in \mathbb{R}^{d \times 1}$ is the $d$-dimensional visual feature vector of the $i$-th image in the test set, $l_i^{(u)} \in \{1, ..., q\}$ is the unknown label of $\mathbf{x}_i^{(u)}$ according to $\mathcal{U}$, $\mathbf{y}_{l_i^{(u)}}^{(u)}$ is the unknown semantic representation of $\mathbf{x}_i^{(u)}$, and $N_u$ denotes the total number of unlabelled images. The goal of zero-shot learning is to predict the labels of test images by learning a classifier $f : \mathcal{X}_u \rightarrow \mathcal{U}$, where $\mathcal{X}_u = \{\mathbf{x}_i^{(u)} : i = 1, ..., N_u\}$.

### 3.2  Model Formulation

As we have mentioned, our DIPL model integrates both forward and reverse projections for ZSL, so that a feature vector representing the visual appearance of an object will be projected into a semantic space and back to reconstruct itself. Such a self-reconstruction task can help narrow the domain gap (see more explanation below). Specifically, assuming that the forward and reverse projections have the same importance for ZSL, our DIPL model solves the following optimization problem:

$$\min_{\mathbf{W}} \left\{ \sum_{i=1}^{N_s} \left( \|\mathbf{W}^T \mathbf{x}_i^{(s)} - \mathbf{y}_{l_i^{(s)}}^{(s)}\|_2^2 + \|\mathbf{x}_i^{(s)} - \mathbf{W}\mathbf{y}_{l_i^{(s)}}^{(s)}\|_2^2 \right) + \lambda \|\mathbf{W}\|_F^2 \right.$$
$$\left. + \gamma \sum_{i=1}^{N_u} \min_j \left( \|\mathbf{W}^T \mathbf{x}_i^{(u)} - \mathbf{y}_j^{(u)}\|_2^2 + \|\mathbf{x}_i^{(u)} - \mathbf{W}\mathbf{y}_j^{(u)}\|_2^2 \right) \right\}, \tag{1}$$

where $\mathbf{W} \in \mathbb{R}^{d \times k}$ is a projection matrix from the semantic space to the feature space, and $\lambda, \gamma$ are the regularization parameters. The first term of Eq. (1) integrates the losses of the forward and reverse projections between the feature and semantic representations of the seen class samples.

Our motivation can be explained as follows: (1) Adding the losses of the forward and reverse projections imposes a self-reconstruction constraint on our regression model, similar to that used in autoencoder [5, 24]. This is motivated by the fact that adding an autoencoder style self-reconstruction task can improve the model generalization ability as demonstrated in many other problems [30, 3]. In our ZSL problem, this improved generalization ability makes the learned regression model more applicable to the unseen class domain. (2) We also apply the similar loss function to the unlabelled unseen class samples (i.e. the third term of Eq. (1)), so that for each unseen class image, its nearest unseen class is found and their distance in the embedding space is minimized. This induces a transductive learning formulation into our model that enables the exploitation of the unlabelled unseen class data for narrowing down the domain gap. In summary, the combination of the auxiliary self-reconstruction task and transductive learning formulation distinguishes our model from existing ones and explains its superior performance. In particular, the generalized ZSL results in Table 3(b) show that our model produces the smallest gap between the seen and unseen class accuracies whilst existing ZSL models heavily favor one over the other. More importantly, although our model only takes a simple linear formulation, it is *clearly shown to outperform* existing nonlinear autoencoder-based ZSL models (including transductive ones) [59, 38] (see Table 2).

### 3.3 Optimization

Since the third term of the objective function in Eq. (1) is denoted as a sum of minimums, it is non-trivial to solve the optimization problem in Eq. (1). In the following, we will formulate our solver as a novel iterative gradient-based algorithm. Note that the contentional alternating optimization algorithms (like k-means) have been employed for solving this type of min-min optimization problems in many existing transductive ZSL models [51, 58, 65]. However, our optimization algorithm is clearly shown to yield better results than these contentional optimization algorithms (see Table 2). This is also the place where our main contribution lies.

Given the projection matrix $\mathbf{W}^{(t)}$ at iteration $t$ during model learning, we define the loss function $\mathbf{f}_i^{(t)} = [f_{i1}^{(t)}, ..., f_{iq}^{(t)}]^T$ for the test image $\mathbf{x}_i^{(u)}$ ($i = 1, ..., N_u$), where $f_{ij}^{(t)} = \|\mathbf{W}^{(t)T}\mathbf{x}_i^{(u)} - \mathbf{y}_j^{(u)}\|_2^2 + \|\mathbf{x}_i^{(u)} - \mathbf{W}^{(t)}\mathbf{y}_j^{(u)}\|_2^2$ ($j = 1, ..., q$). For the minimum function $\min \mathbf{f}_i^{(t)}$, we define its gradient $\eta_i^{(t)} = [\eta_{i1}^{(t)}, ..., \eta_{iq}^{(t)}]^T$ with respect to $\mathbf{f}_i^{(t)}$ as follows:

$$\eta_{ij}^{(t)} = \begin{cases} 1/n_i^{(t)} & , \text{ if } f_{ij}^{(t)} = \min \mathbf{f}_i^{(t)} \\ 0 & , \text{ otherwise} \end{cases}, \tag{2}$$

where $n_i^{(t)}$ is the number of $f_{ij}^{(t)}$ ($j = 1, ..., q$) being equal to $\min \mathbf{f}_i^{(t)}$. Taking the Taylor expansion, we have the following approximation:

$$\min_j \left( \|\mathbf{W}^{(t+1)T}\mathbf{x}_i^{(u)} - \mathbf{y}_j^{(u)}\|_2^2 + \|\mathbf{x}_i^{(u)} - \mathbf{W}^{(t+1)}\mathbf{y}_j^{(u)}\|_2^2 \right)$$
$$= \min \mathbf{f}_i^{(t+1)} \approx \min \mathbf{f}_i^{(t)} + \eta_i^{(t)T}(\mathbf{f}_i^{(t+1)} - \mathbf{f}_i^{(t)}) = \eta_i^{(t)T}\mathbf{f}_i^{(t+1)}. \tag{3}$$

The objective function in Eq. (1) at iteration $t + 1$ can be estimated as:

$$\mathcal{F}(\mathbf{W}^{(t+1)}) = \sum_{i=1}^{N_s} \left( \|\mathbf{W}^{(t+1)T}\mathbf{x}_i^{(s)} - \mathbf{y}_{l_i^{(s)}}^{(s)}\|_2^2 + \|\mathbf{x}_i^{(s)} - \mathbf{W}^{(t+1)}\mathbf{y}_{l_i^{(s)}}^{(s)}\|_2^2 \right)$$
$$+ \gamma \sum_{i=1}^{N_u} \eta_i^{(t)T}\mathbf{f}_i^{(t+1)} + \lambda\|\mathbf{W}^{(t+1)}\|_F^2. \tag{4}$$

Let $\frac{\partial \mathcal{F}(\mathbf{W}^{(t+1)})}{\partial \mathbf{W}^{(t+1)}} = 0$, we obtain a linear equation as follows:

$$\mathbf{A}^{(t)}\mathbf{W}^{(t+1)} + \mathbf{W}^{(t+1)}\mathbf{B}^{(t)} = \mathbf{C}^{(t)}, \tag{5}$$

where $\mathbf{A}^{(t)} = \sum_{i=1}^{N_s} \mathbf{x}_i^{(s)}\mathbf{x}_i^{(s)T} + \gamma \sum_{i=1}^{N_u} \mathbf{x}_i^{(u)}\mathbf{x}_i^{(u)T} + \lambda I$, $\mathbf{B}^{(t)} = \sum_{i=1}^{N_s} \mathbf{y}_{l_i^{(s)}}^{(s)}\mathbf{y}_{l_i^{(s)}}^{(s)T} + \gamma \sum_{i=1}^{N_u}\sum_{j=1}^q \eta_{ij}^{(t)}\mathbf{y}_j^{(u)}\mathbf{y}_j^{(u)T}$, and $\mathbf{C}^{(t)} = 2\sum_{i=1}^{N_s} \mathbf{x}_i^{(s)}\mathbf{y}_{l_i^{(s)}}^{(s)T} + 2\gamma \sum_{i=1}^{N_u}\sum_{j=1}^q \eta_{ij}^{(t)}\mathbf{x}_i^{(u)}\mathbf{y}_j^{(u)T}$. Let $\alpha_t = \gamma/(1+\gamma) \in (0, 1)$ and $\beta = \lambda/(1+\gamma)$. In this paper, we empirically set $\alpha_t = 0.99^t\alpha$ ($\alpha_0 =$

---
**Algorithm 1** Domain-Invariant Projection Learning
---
**Input:** training and test sets $\mathcal{D}_s, \mathcal{X}_u$; semantic prototypes $\mathbf{Y}_s, \mathbf{Y}_u$; parameter $\alpha$
**Output:** $\mathbf{W}^*$
1. Initialize $\mathbf{W}^{(0)}$ with our DIPL model ($\alpha = 0$) at $t = 0$;
**repeat**
   2. Set $\alpha_t = 0.99^t \alpha$;
   3. With the learned projection matrix $\mathbf{W}^{(t)}$, compute the gradient $\eta_{ij}^{(t)}$ with Eq. (2);
   4. Compute $\widehat{\mathbf{A}}^{(t)}, \widehat{\mathbf{B}}^{(t)}$, and $\widehat{\mathbf{C}}^{(t)}$ with Eqs. (6)–(8), and update $\mathbf{W}^{(t+1)}$ by solving Eq. (9);
   5. Set $t = t + 1$;
**until** *a stopping criterion is met*
6. $\mathbf{W}^* = \mathbf{W}^{(t)}$.
---

$\alpha \in (0, 1))$ and $\beta = 0.01$ in all experiments. We thus have:

$$\widehat{\mathbf{A}}^{(t)} = (1 - \alpha_t) \sum_{i=1}^{N_s} \mathbf{x}_i^{(s)} \mathbf{x}_i^{(s)T} + \alpha_t \sum_{i=1}^{N_u} \mathbf{x}_i^{(u)} \mathbf{x}_i^{(u)T} + \beta I, \tag{6}$$

$$\widehat{\mathbf{B}}^{(t)} = (1 - \alpha_t) \sum_{i=1}^{N_s} \mathbf{y}_{l_i^{(s)}}^{(s)} \mathbf{y}_{l_i^{(s)}}^{(s)T} + \alpha_t \sum_{i=1}^{N_u} \sum_{j=1}^{q} \eta_{ij}^{(t)} \mathbf{y}_j^{(u)} \mathbf{y}_j^{(u)T}, \tag{7}$$

$$\widehat{\mathbf{C}}^{(t)} = 2(1 - \alpha_t) \sum_{i=1}^{N_s} \mathbf{x}_i^{(s)} \mathbf{y}_{l_i^{(s)}}^{(s)T} + 2\alpha_t \sum_{i=1}^{N_u} \sum_{j=1}^{q} \eta_{ij}^{(t)} \mathbf{x}_i^{(u)} \mathbf{y}_j^{(u)T}. \tag{8}$$

The linear equation in Eq. (5) is then reformulated as follows:

$$\widehat{\mathbf{A}}^{(t)} \mathbf{W}^{(t+1)} + \mathbf{W}^{(t+1)} \widehat{\mathbf{B}}^{(t)} = \widehat{\mathbf{C}}^{(t)}, \tag{9}$$

which is a Sylvester equation and it can be solved efficiently by the Bartels-Stewart algorithm [6].

The DIPL algorithm is given in Algorithm 1, with rigorous theoretic algorithm analysis in the suppl. material. Note that any ZSL model can be used to obtain the initial projection matrix $\mathbf{W}^{(0)}$. In this paper, we choose our DIPL model with $\alpha = 0$ for this initialization. Once learned, given the optimal projection matrix $\mathbf{W}^*$ found by our DIPL algorithm, we predict the label of a test image $\mathbf{x}_i^{(u)}$ as: $l_i^{(u)} = \arg\min_j \left( \|\mathbf{W}^{*T} \mathbf{x}_i^{(u)} - \mathbf{y}_j^{(u)}\|_2^2 + \|\mathbf{x}_i^{(u)} - \mathbf{W}^* \mathbf{y}_j^{(u)}\|_2^2 \right)$.

We provide the time complexity analysis for Algorithm 1 as follows. The computation of $[\eta_{ij}^{(t)}]_{N_u \times q}$, $\widehat{\mathbf{A}}^{(t)}, \widehat{\mathbf{B}}^{(t)}$, and $\widehat{\mathbf{C}}^{(t)}$ has a time complexity of $O(qN_u)$, $O(d^2(N_s + N_u))$, $O(k^2 N_s + k^2 N_u)$, and $O(dkN_s + dkN_u)$, respectively. Here, the sparsity of $[\eta_{ij}^{(t)}]$ is used to reduce the cost of computing $\widehat{\mathbf{B}}^{(t)}$ and $\widehat{\mathbf{C}}^{(t)}$. Moreover, given $\widehat{\mathbf{A}}^{(t)} \in \mathbb{R}^{d \times d}$ and $\widehat{\mathbf{B}}^{(t)} \in \mathbb{R}^{k \times k}$, the time complexity of solving Eq. (9) is $O(d^3 + k^3)$. To sum up, one iteration has a linear time complexity of $O(qN_u + (d^2 + dk + k^2)(N_s + N_u))$ $(d, k, q \ll (N_s + N_u))$ with respect to the data size $N_s + N_u$. Since Algorithm 1 is shown to converge very quickly ($t \le 5$), it is efficient even for large-scale ZSL problems.

### 3.4 ZSL with Superclasses

We finally apply our DIPL algorithm to ZSL with superclasses. This is motivated by the fact that there exist unseen/seen classes that fall into the same superclass, i.e., the unseen class samples become 'seen' at the superclass level and thus easier to recognize. Specifically, our DIPL model is employed for ZSL with superclasses as follows: 1) Group all unseen and seen class prototypes $[\mathbf{Y}_s, \mathbf{Y}_u]$ into $r$ clusters by k-means clustering and represent the superclass prototypes with the cluster centers $\mathbf{Z} = [\mathbf{z}_1, ..., \mathbf{z}_r]$; 2) Run our DIPL algorithm over superclasses by replacing the original semantic prototypes $[\mathbf{Y}_s, \mathbf{Y}_u]$ by the superclass prototypes $\mathbf{Z}$; 3) Predict the top 5 superclass labels of each unlabelled unseen sample $\mathbf{x}_i^{(u)}$ and then generate the set of the most possible unseen class labels $\mathcal{N}(\mathbf{x}_i^{(u)})$ for $\mathbf{x}_i^{(u)}$ according to the k-means clustering results; 4) Run our DIPL algorithm over the original semantic prototypes $[\mathbf{Y}_s, \mathbf{Y}_u]$ by computing $\eta_{ij}^{(t)}$ with the constraint $j \in \mathcal{N}(\mathbf{x}_i^{(u)})$.

Table 1: Five benchmark datasets used for performance evaluation. Notations: 'SS' – semantic space, 'SS-D' – the dimension of semantic space, 'A' – attribute, and 'W' – word vector. The two splits of the SUN dataset are separated by '|'.

| Dataset | # images | SS | SS-D | # seen/unseen |
|---------|----------|-----|------|---------------|
| AwA | 30,475 | A | 85 | 40/10 |
| CUB | 11,788 | A | 312 | 150/50 |
| aPY | 15,339 | A | 64 | 20/12 |
| SUN | 14,340 | A | 102 | 707/10\|645/72 |
| ImNet | 218,000 | W | 1,000 | 1,000/360 |

## 4 Experiments

### 4.1 Datasets and Settings

**Datasets**. Five widely-used benchmark datasets are selected in this paper. Four of them are of medium-size: Animals with Attributes (AwA) [25], CUB-200-2011 Birds (CUB) [56], aPascal&Yahoo (aPY) [13], and SUN Attribute (SUN) [41]. One large-scale dataset is ILSVRC2012/2010 [47] (ImNet), where the 1,000 classes of ILSVRC2012 are used as seen classes and 360 classes of ILSVRC2010 (not included in ILSVRC2012) are used as unseen classes, as in [16]. The details of these benchmark datasets are given in Table 1.

**Semantic Spaces**. Two types of semantic spaces are considered for ZSL: attributes are employed to form the semantic space for the four medium-scale datasets, while word vectors are used as semantic representation for the large-scale ImNet dataset. In this paper, we train a skip-gram text model on a corpus of 4.6M Wikipedia documents to obtain the word2vec [37] word vectors.

**Visual Spaces**. All recent ZSL models use the visual features extracted by CNN models [53, 55, 19], which are pre-trained on the 1K classes in ILSVRC 2012 [47]. In this paper, we extract the visual features with pre-trained GoogLeNet [55]. Note that the same visual features (GoogLeNet) are used for most compared methods throughout this paper. The only exception is Table 2, where although most results were obtained with GoogLeNet features, a number of more recent ZSL models used VGG19 [53] and ResNet101 [19] features. Without source code of these models, we cannot report their results with the same GoogLeNet features. However, as demonstrated in [28], the VGG19 and ResNet101 features typically lead to better performance in the ZSL task than the GoogLeNet features. Since our model does not use stronger features, the comparisons in Table 2 are still fair.

**ZSL Settings**. (1) **Standard ZSL**: This setting is widely used in previous works [2, 44]. The seen/unseen class splits of the five datasets are presented in Table 1. (2) **Pure ZSL**: A new 'pure' ZSL setting [61, 29] is recently proposed to overcome the weakness of the standard setting. More concretely, most recent ZSL models extract the visual features using ImageNet ILSVRC2012 1K classes pretrained CNN models, but the unseen classes in the standard splits overlap with the 1K ImageNet classes. The zero-shot rule is thus violated. Under the pure setting, the overlapped ImageNet classes are removed from the test set of unseen classes for the new benchmark ZSL dataset splits. (3) **Generalized ZSL**: The third ZSL setting that emerges recently [43, 7] is the generalized setting under which the test set contains data samples from both seen and unseen classes. This setting is clearly more reflective of real-world application scenarios.

**Evaluation Metrics**. (1) **Standard and Pure ZSL**: For the four medium-scale datasets, we compute the multi-way classification accuracy as in previous works. For the large-scale ImNet dataset, the flat hit@5 accuracy is computed over all test samples as in [16]. (2) **Generalized ZSL**: Three metrics are defined as: 1) $acc_s$ – the accuracy of classifying the data samples from the seen classes to all the classes (both seen and unseen); 2) $acc_u$ – the accuracy of classifying the data samples from the unseen classes to all the classes; 3) HM – the harmonic mean of $acc_s$ and $acc_u$.

**Parameter Settings**. Our full DIPL model (including superclasses) has only two free parameters to tune: $\alpha \in (0, 1)$ (see Step 2 in Algorithm 1) and $r$ (the number of superclasses used in Sec. 3.4). As in [50, 24], the parameters are selected by class-wise cross-validation on the training set.

**Compared Methods**. In this paper, a wide range of existing ZSL models are selected for performance comparison. Under each ZSL setting, we focus on the recent and representative ZSL models that have achieved the state-of-the-art results.

Table 2: Comparative accuracies (%) under the standard ZSL setting. For SUN, the results are obtained for the 707/10 and 645/72 splits, separated by '|'. For ImNet, the hit@5 accuracy is used for evaluation. Visual features: G – GoogLeNet [55]; V – VGG19 [53]; R – ResNet101 [19].

| Model | Features | Trans.? | AwA | CUB | aPY | SUN | | ImNet |
|---|---|---|---|---|---|---|---|---|
| RPL [50] | G | N | 80.4 | 52.4 | 48.8 | 84.5 | – | – |
| SSE [67] | V | N | 76.3 | 30.4 | 46.2 | 82.5 | – | – |
| SJE [2] | G | N | 73.9 | 51.7 | – | – | 56.1 | – |
| JLSE [68] | V | N | 80.5 | 42.1 | 50.4 | 83.8 | – | – |
| SynC [8] | G | N | 72.9 | 54.7 | – | – | 62.7 | – |
| SAE [24] | G | N | 84.7 | 61.4 | 55.4 | 91.5 | 65.2 | 27.2 |
| LAD [21] | V | N | 82.5 | 56.6 | 53.7 | 85.0 | – | – |
| EXEM [9] | G | N | 77.2 | 59.8 | – | – | 69.6 | – |
| SCoRe [39] | V | N | 82.8 | 59.5 | – | – | – | – |
| LESD [11] | V/G | N | 82.8 | 56.2 | 58.8 | 88.3 | – | – |
| CVA [38] | V/R | N | 85.8 | 54.3 | – | 88.5 | – | 24.7 |
| f-CLSWGAN [60] | R | N | 69.9 | 61.5 | – | – | 62.1 | – |
| SS-Voc [16] | V | Y | 78.3 | – | – | – | – | 16.8 |
| SP-ZSR [69] | V | Y | 92.1 | 55.3 | 69.7 | 89.5 | – | – |
| SSZSL [51] | V | Y | 88.6 | 58.8 | 49.9 | 86.2 | – | – |
| DSRL [64] | V | Y | 87.2 | 57.1 | 56.3 | 85.4 | – | – |
| TSTD [65] | V | Y | 90.3 | 58.2 | – | – | – | – |
| BiDiLEL [58] | V/G | Y | 95.0 | 62.8 | – | – | – | – |
| DMaP [28] | V+G+R | Y | 90.5 | 67.7 | – | – | – | – |
| VZSL [59] | V | Y | 94.8 | 66.5 | – | 87.8 | – | 23.1 |
| Full DIPL (our) | G | Y | **96.1** | **68.2** | **87.8** | **93.5** | **70.0** | **31.7** |

Table 3: (a) Comparative accuracies (%) under the pure ZSL setting (as in [61, 29]). (b) Comparative results (%) of generalized ZSL (as in [10]). For the SUN dataset, only the 645/72 split is used.

| Model | AwA | CUB | aPY | SUN |
|---|---|---|---|---|
| DeViSE [14] | 54.2 | 52.0 | 39.8 | 56.5 |
| ConSE [40] | 45.6 | 34.3 | 26.9 | 38.8 |
| SSE [67] | 60.1 | 43.9 | 34.0 | 51.5 |
| SJE [2] | 65.6 | 53.9 | 32.9 | 53.7 |
| ALE [1] | 59.9 | 54.9 | 39.7 | 58.1 |
| SynC [8] | 54.0 | 55.6 | 23.9 | 56.3 |
| CLN+KRR [29] | 68.2 | 58.1 | 44.8 | 60.0 |
| Full DIPL (our) | **85.6** | **65.4** | **69.6** | **67.9** |

(a)

| Model | AwA | | | CUB | | |
|---|---|---|---|---|---|---|
| | $acc_s$ | $acc_u$ | HM | $acc_s$ | $acc_u$ | HM |
| DAP [25] | 77.9 | 2.4 | 4.7 | 55.1 | 4.0 | 7.5 |
| IAP [25] | 76.8 | 1.7 | 3.3 | 69.4 | 1.0 | 2.0 |
| ConSE [40] | 75.9 | 9.5 | 16.9 | 69.9 | 1.8 | 3.5 |
| APD [43] | 43.2 | 61.7 | 50.8 | 23.4 | 39.9 | 29.5 |
| GAN [7] | 81.3 | 32.3 | 46.2 | **72.0** | 26.9 | 39.2 |
| SAE [24] | 67.6 | 43.3 | 52.8 | 36.1 | 28.0 | 31.5 |
| Full DIPL (our) | **83.7** | **68.9** | **75.6** | 44.8 | **41.7** | **43.2** |

(b)

## 4.2 Comparative Results

**Standard ZSL**. The comparative results under the standard ZSL setting are shown in Table 2. For comprehensive comparison, both transductive and non-transductive state-of-the-art ZSL models are included. It can be seen that: (1) Our model performs the best on all five datasets, validating that the combination of domain-invariant feature self-reconstruction and superclasses shared across domain alignment is indeed effective for learning domain-invariant projection. (2) For the four medium-scale datasets, the improvements obtained by our model over the strongest competitor range from 0.4% to 18.1%. This actually creates new baselines in the area of ZSL, given that most of the compared models take far more complicated nonlinear formulations and some of them even combine two or more feature/semantic spaces. (3) For the large-scale ImNet dataset, our model achieves a 4.5% improvement over the state-of-the-art SAE [24], showing its scalability to large-scale problems.

**Pure ZSL**. Taking the same 'pure' ZSL setting as in [61, 29], we remove the overlapped ImageNet ILSVRC2012 1K classes from the test set of unseen classes for the four medium-scale datasets. The comparative results in Table 3(a) show that, as expected, under this stricter ZSL setting, all ZSL models suffer from performance degradation. However, the performance of our model drops the least among all ZSL models, and the improvement over the strongest competitor becomes more significant for each of the four datasets. This provides further evidence that our model tends to learn a domain-invariant projection even under this stricter ZSL setting.

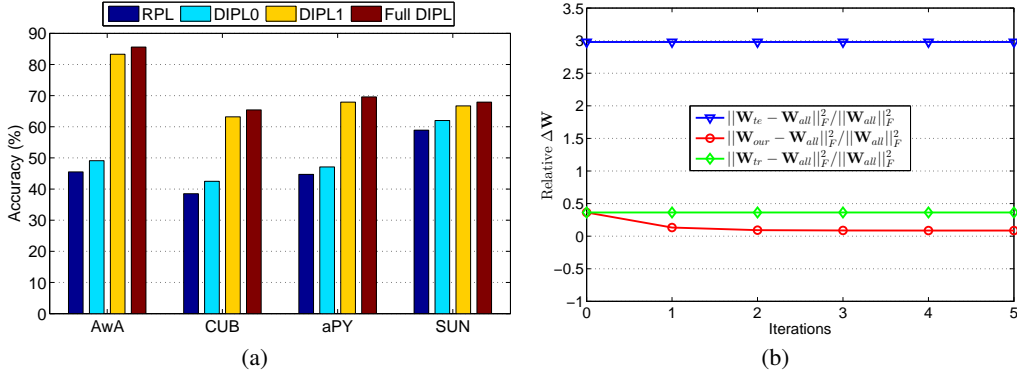

Figure 1: (a) Ablation study results on the four medium-scale datasets under the pure ZSL setting. (b) Convergence analysis of our DIPL algorithm on the CUB dataset under the pure ZSL setting.

**Generalized ZSL**. We follow the same generalized ZSL setting of [10]. Specifically, we hold out 20% of the data samples from the seen classes and mix them with the data samples from the unseen classes. The comparative results on AwA and CUB are presented in Table 3(b), where our model is compared with six other ZSL alternatives. We have the following observations: (1) Different ZSL models have a different trade-off between the seen and unseen class accuracies, and the overall performance is thus best measured by HM. (2) Our model clearly performs the best over the two datasets, and its advantage over other competitors is even more significant for this more challenging setting. (3) Our model produces the smallest gap between the seen and unseen class accuracies whilst existing ZSL models heavily favor one over the other. This means that our model has the strongest generalization ability under this more realistic ZSL setting.

## 4.3 Further Evaluations

**Ablation Study**. Our full DIPL model can be simplified as follows: (1) When the superclasses are not used for ZSL, our full DIPL model degrades to the original DIPL model proposed in Sec. 3.3, denoted as DIPL1; (2) For $\alpha = 0$, the DIPL1 model further degrades to an inductive ZSL model (including both forward and reverse projections), denoted as DIPL0; (3) When the forward projection is not considered for ZSL, the DIPL0 model finally degrades to the original reverse projection learning model [50], denoted as RPL. To evaluate the contributions of the main components of our full DIPL model, we compare it with the simplified versions RPL, DIPL0, and DIPL1 under the same pure ZSL setting. The ablation study results in Figure 1(a) show that: (1) The transductive learning induced by our DIPL1 model yields significant improvements (see DIPL1 vs. DIPL0), ranging from 5% to 30%. (2) Our enhanced ZSL method using superclasses achieves about 1–2% gains (see Full DIPL vs. DIPL1), validating its effectiveness. This is still very impressive since the DIPL1 model has already achieved state-of-the-art results. More results of ZSL with superclasses are provided in Table 4. (3) The combination of both forward and reverse projections is also important for ZSL (see DIPL0 vs. RPL), resulting in 2–4% improvements.

**Convergence Analysis**. To provide more convergence analysis for Algorithm 1, we define three baseline projection matrices based on the DIPL0 model: 1) $\mathbf{W}_{all}$ – learned by DIPL0 using the whole dataset (all are labelled); 2) $\mathbf{W}_{tr}$ – learned by DIPL0 only using the training set; 3) $\mathbf{W}_{te}$ – learned by DIPL0 only using the test set (but labelled). Let $\mathbf{W}_{our}$ be learned by our DIPL model using the test set (unlabelled) and the training set. We can directly compare $\mathbf{W}_{our}$, $\mathbf{W}_{tr}$, and $\mathbf{W}_{te}$ to $\mathbf{W}_{all}$ by computing the matrix distances among these matrices. Note that $\mathbf{W}_{all}$ is considered to be the best possible projection matrix (upper bound). The results in Figure 1(b) show that: (1) Our DIPL algorithm converges very quickly ($\leq 5$ iterations). (2) $\mathbf{W}_{our}$ gets closer to $\mathbf{W}_{all}$ with more iterations and it is the closest to $\mathbf{W}_{all}$ at convergence, i.e., our model can narrow the domain gap by not overfitting to the training domain.

**Qualitative Results**. We present the qualitative results of superclass generation on the ImNet dataset in Table 4. The superclasses are generated by k-means clustering (with $r = 500$ clusters) on all seen/unseen class prototypes. We have the following observations: (1) There indeed exist

Table 4: Examples of the superclasses generated by k-means clustering on the ImNet dataset.

| Superclasses | Seen/unseen classes within a superclass |
|:---:|:---|
| ID: 1 | seen: unicycle; unseen: hard hat |
| ID: 2 | seen: freight car; unseen: ferris wheel |
| ID: 3 | seen: hair slide; unseen: nail polish |
| ID: 4 | seen: ox; seen: bison |
| ID: 5 | unseen: coffee bean; unseen: Arabian coffee; unseen: cacao |

superclasses that consist of semantically-related seen and unseen classes, which means that the unseen class samples can become 'seen' in ZSL with superclasses and thus easier to recognize. (2) When only unseen (or only seen) classes are included in a superclass, they are also semantically related, which can be used as the context to improve the performance of label prediction.

## 4.4 Discussions

We have reformulated our model with the soft assignment (e.g. using softmax loss), but the results are clearly worse. One of the possible reasons is in the solver: with the min-min formulation, the problem can be solved explicitly using a linear equation at each iteration (see Eq. (9)). In contrast, the nonlinear min-softmax problem is harder to solve and the standard gradient-based solver does not have the nice convergence property as in the min-min formulation.

Among the five datasets used in our experiments, our DIPL model is shown to be only able to achieve small improvements on the CUB dataset. Unlike the other four datasets, CUB is much more fine-grained – all classes are sub-species of birds. As a result, the unseen classes of CUB are very similar. It thus becomes hard for our DIPL model to find the best unseen class label for an unlabelled unseen class sample during training. This shortcoming can potentially be overcome by generalized competitive learning [33, 52, 35, 32]: examining the second most likely unseen class label and forcing the projection to distinguish the best and second best ones – essentially pushing the unseen classes further away to each other after projection.

Note that our DIPL model is essentially a bidirectional one, with an autoencoder style self-reconstruction task involved. Although only evaluated in the area of ZSL, our bidirectional model can be applied to other problem settings where a mapping between a feature and a semantic space is required. For example, in our ongoing research, our model has been generalized to social image classification [36] (where social tags form the semantic space) and cross-modal image retrieval [34] (where texts form the semantic space). In both, we find that a bidirectional model is clearly better than a one-direction one. We also notice that recently bidirectional models have found success in problems involving identity space, e.g., face recognition and person re-identification [57].

## 5 Conclusion

In this paper, we have proposed a domain-invariant projection learning (DIPL) model for zero-shot recognition. A novel iterative algorithm has been developed for model optimization, followed by rigorous theoretic algorithm analysis. Our model has also been extended to ZSL with superclasses. Extensive experiments on five benchmark datasets show that our DIPL model yields state-of-the-art results under all three ZSL settings. It is worth pointing out that the proposed optimization algorithm is by no means restricted to the ZSL problem – many other vision problems need to deal with a min-min problem and thus our gradient-based formulation can be induced similarly. Our current efforts thus include its generalization to solve a wider range of vision problems (e.g. social image classification, cross-modal image retrieval, and person re-identification).

## Acknowledgements

This work was partially supported by National Natural Science Foundation of China (61573363), the Fundamental Research Funds for the Central Universities and the Research Funds of Renmin University of China (15XNLQ01), and European Research Council FP7 Project SUNNY (313243).

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
