[Supplementary Material]

# Domain-Invariant Projection Learning for Zero-Shot Recognition (Supplementary Material)

**An Zhao**[1],• **Mingyu Ding**[1],• **Jiechao Guan**[1],• **Zhiwu Lu**[1,*] **Tao Xiang**[2,3] **Ji-Rong Wen**[1]

[1]Beijing Key Laboratory of Big Data Management and Analysis Methods
School of Information, Renmin University of China, Beijing 100872, China
[2]School of EECS, Queen Mary University of London, London E1 4NS, U.K.
[3]Samsung AI Centre, Cambridge, U.K.
zhiwu.lu@gmail.com    t.xiang@qmul.ac.uk
• Equal contribution    * Corresponding author

## 1  Algorithm Analysis

In this section, we provide a rigorous analysis on the properties and behaviours of the optimisation algorithm formulated in Section 3.3 (Algorithm 1) of the main paper.

**Proposition 1** *Eq. (9) has and only has one solution.*

*Proof.* Because $\beta > 0$, $\widehat{\mathbf{A}}^{(t)}$ is positive definite, i.e., $\lambda_i^a \geq \beta > 0$ $(i = 1, ..., d)$. Similarly, $\widehat{\mathbf{B}}^{(t)}$ is positive semidefinite, i.e., all of its eigenvalues satisfy: $\lambda_j^b \geq 0$ $(j = 1, ..., k)$. The eigenvalue decompositions of $\widehat{\mathbf{A}}^{(t)}$ and $\widehat{\mathbf{B}}^{(t)}$ are denoted as: $\widehat{\mathbf{A}}^{(t)} = V\Sigma_A V^T$ ($\Sigma_A = \mathrm{diag}\{\lambda_1^a, ..., \lambda_d^a\}, V^T V = I$), $\widehat{\mathbf{B}}^{(t)} = U\Sigma_B U^T$ ($\Sigma_B = \mathrm{diag}\{\lambda_1^b, ..., \lambda_k^b\}, U^T U = I$). Therefore, Eq. (9) is reformulated as: $\Sigma_A V^T \mathbf{W}^{(t+1)} U + V^T \mathbf{W}^{(t+1)} U\Sigma_B = V^T \widehat{\mathbf{C}}^{(t)} U$. Let $\overline{\mathbf{W}} = V^T \mathbf{W}^{(t+1)} U$ and $\overline{\mathbf{C}} = V^T \widehat{\mathbf{C}}^{(t)} U$. We have: $\Sigma_A \overline{\mathbf{W}} + \overline{\mathbf{W}}\Sigma_B = \overline{\mathbf{C}}$, i.e., $(\lambda_i^a + \lambda_j^b)\overline{w}_{ij} = \overline{c}_{ij}$ $(i = 1, ..., d; \ j = 1, ..., k)$. Since $\lambda_i^a + \lambda_j^b > 0$ and $\overline{\mathbf{W}} = V^T \mathbf{W}^{(t+1)} U$, Eq. (9) has and only has one solution. $\qquad\square$

**Proposition 2** *Given $\Delta\mathbf{W}^{(t)} = \mathbf{W}^{(t+1)} - \mathbf{W}^{(t)}$, we have: $\lim_{t\to+\infty} \|\Delta\mathbf{W}^{(t)}\|_F^2 = 0$, i.e., Algorithm 1 is a convergent iterative algorithm.*

*Proof.* Without loss of generality, we normalize all of $\|\mathbf{x}_i^{(s)}\|_2^2$, $\|\mathbf{x}_i^{(u)}\|_2^2$, $\|\mathbf{y}_j^{(s)}\|_2^2$, and $\|\mathbf{y}_j^{(u)}\|_2^2$ to 1 (see Eqs. (6)–(8)). We can easily have: $\|\Delta\widehat{\mathbf{A}}^{(t-1)}\|_F^2 = \|\widehat{\mathbf{A}}^{(t)} - \widehat{\mathbf{A}}^{(t-1)}\|_F^2 \leq \alpha_{t-1}\Delta\widehat{\mathbf{A}}$, $\|\Delta\widehat{\mathbf{B}}^{(t-1)}\|_F^2 = \|\widehat{\mathbf{B}}^{(t)} - \widehat{\mathbf{B}}^{(t-1)}\|_F^2 \leq \alpha_{t-1}\Delta\widehat{\mathbf{B}}$, and $\|\Delta\widehat{\mathbf{C}}^{(t-1)}\|_F^2 = \|\widehat{\mathbf{C}}^{(t)} - \widehat{\mathbf{C}}^{(t-1)}\|_F^2 \leq \alpha_{t-1}\Delta\widehat{\mathbf{C}}$, where $\Delta\widehat{\mathbf{A}}$, $\Delta\widehat{\mathbf{B}}$, and $\Delta\widehat{\mathbf{C}}$ are all positive constants. Moreover, according to the proof of Prop. 1, we have: $(\lambda_i^a + \lambda_j^b)\overline{w}_{ij} = \overline{c}_{ij}$ $(i = 1, ..., d; \ j = 1, ..., k)$. Given that $\lambda_i^a + \lambda_j^b \geq \beta > 0$, we have: $|\overline{w}_{ij}| \leq |\overline{c}_{ij}|/\beta$. Since $\overline{\mathbf{W}} = V^T \mathbf{W}^{(t+1)} U$ and $\overline{\mathbf{C}} = V^T \widehat{\mathbf{C}}^{(t)} U$, we have: $\|\mathbf{W}^{(t+1)}\|_F^2 \leq \|\widehat{\mathbf{C}}^{(t)}\|_F^2/\beta^2 \leq M_C/\beta^2$, where $M_C$ is a positive constant. By subtracting Eq. (9) at $t - 1$ from Eq. (9) at $t$, we thus obtain: $\widehat{\mathbf{A}}^{(t)}\Delta\mathbf{W}^{(t)} + \Delta\mathbf{W}^{(t)}\widehat{\mathbf{B}}^{(t)} = \Delta\widehat{\mathbf{D}}^{(t-1)}$, where $\Delta\widehat{\mathbf{D}}^{(t-1)} = \Delta\widehat{\mathbf{C}}^{(t-1)} - \Delta\widehat{\mathbf{A}}^{(t-1)}\mathbf{W}^{(t)} - \mathbf{W}^{(t)}\Delta\widehat{\mathbf{B}}^{(t-1)}$. According to the proof that $\|\mathbf{W}^{(t+1)}\|_F^2 \leq \|\widehat{\mathbf{C}}^{(t)}\|_F^2/\beta^2$, we can similarly obtain: $\|\Delta\mathbf{W}^{(t)}\|_F^2 \leq \|\Delta\widehat{\mathbf{D}}^{(t-1)}\|_F^2/\beta^2$. Since $\|\Delta\widehat{\mathbf{D}}^{(t-1)}\|_F^2 \leq \alpha_{t-1}[\Delta\widehat{\mathbf{C}} + (\Delta\widehat{\mathbf{A}} + \Delta\widehat{\mathbf{B}})M_C/\beta^2]$ and $\lim_{t\to+\infty}\alpha_{t-1} = 0$, we have: $\lim_{t\to+\infty}\|\Delta\mathbf{W}^{(t)}\|_F^2 = 0$. $\qquad\square$