[Reviews · NeurIPS 2018]

Reviewer 1



This paper presents an elegant method of zero-shot recognition based on transductive domain-invariant projection learning over both seen and unseen classes. Main theoretical contributions are: (1) the idea of introducing self-reconstruction constraint for unseen classes data as well as seen ones (the third term in eq.1), (2) derivation of an efficient optimization algorithm for this min-min problem, and (3) introducing superclasses to better align two domains. The proposed DIPL method is shown to significantly outperform previous state-of-the-arts on many ZSL tasks. Strengths: 1. The basic formulation (eq.1) is conceptually simple and reasonable, making the overall method beautiful and general. I think the idea can be useful in many related areas of few-resource learning. 2. The optimization algorithm to handle the transductive constraint (the third term in eq.1) is non-trivial and elegantly derived, resulting in a linear time complexity with respect to the data size which is practical in real situations. 3. Experiments are well-organized and convincing enough to prove the effectiveness of each technical contribution of the method. Particularly, transductive constraint is shown to significantly improve the overall performance. Proposed method is shown to consistently outperforms previous methods on various ZSL tasks. Weaknesses: 1. It sounds unfair to say ‘only two free parameters’ (l. 248) considering another parameter beta is set to 0.01 empirically (l. 167). Considering that beta=lambda/(1+gamma), it should be also data dependent. UPDATES: My score remains still. I think it's a good paper. Sections 3.3 and 3.4 are bit dense and unfriendly, so maybe it would be a good idea to move some of them to supplementary. Also, the paper would be more perfect if authors the results on ImageNet and SUN are reported in the main content.

Reviewer 2



Summary This paper proposes a novel formulation for transductive zero-shot learning. The main idea is to enforce bidirectional reconstruction constraints between visual and semantic features on both labeled data from seen classes and unlabeled data from unseen classes. The term for unlabeled data leads to a min-min optimization problem and a novel solver is further proposed. They also compute the superclasses by clustering to enforce more constraints on possible unseen class labels. The experimental results show significant improvement over the state-of-the-art. Pros -The proposed approach is novel and interesting -This paper is well-written and easy to follow. -The experiments are convincing and show significant improvement over the state-of-the-art. Cons -The generalized zero-shot learning results on SUN are missing. -The large-scale experiments in the supplementary material on ImageNet are superficial. It would be much better if it can be evaluated on 2-hops, 3-hops and generalized zero-shot learning on ImageNet. Additional comments -Line 267: The author says that "we follow the same generalized ZSL setting of [10]". But it seems that this paper is following [52] by using HM instead of using AUSUC like [10]. -How many superclasses are formed? Is it sensitive to the performance?

Reviewer 3



The authors present an algorithm for zero-shot learning based on learning a linear projection between the features of a pre-trained convnet and a semantic space in which to do nearest neighbors classification. Overall, I found the exposition of the paper very confusing, and I am still struggling to understand the exact set-up in the experiments and theorems after several re-readings. I'm not 100% certain from the paper what the authors are actually doing and what they have available as training data. In particular, the authors denote the test images by D_u, which they are sure to point out are unlabeled. They then define l_i^{(u)} to be the label of test point x_i^{(u)} - how can the label be included in a set of test images that is unlabeled? What information from D_u is provided when learning the projection matrix W, and what is held out until test time? What is the algorithm used to map the projection of a feature to a class? I *think* what is happening is that the semantic features y_j for each unseen class j are provided at training time, but not the label for which class each test image belongs to, but this is not well-explained. It is also not clear to me how these vectors y_j are chosen in the experiments. I *think* that the datasets they work with have pre-defined attribute vectors for every class - providing a concrete example of such an attribute vector would help explain the source of these y_j terms. In the loss function in Eq 1, I am not clear on why W and W^T are used for the forward/backward projection from visual features to semantic features, rather than W and its Moore-Penrose pseudoinverse. Is the learned W approximately orthogonal? Line 152: "contentional" is not a word. I am not sure what the authors are trying to say here. Section 3.3: This section is especially confusing. I suppose the confusion stems from the fact that f_i^{(t)} is a vector, and you are doing an optimization over elements of the vector, that is, just taking the minimum element. Not to mention some of the math seems wrong - you define the gradient of the minimum of the element of the vector to be 1/number of elements with that value. The minimum is a piecewise-linear function, and if multiple elements of the vector have the same (minimum) value, the minimum does not have a well-defined gradient at that point and instead has a *sub*gradient, which is a set of directions that fall below the function. Section 3.4: This section is far too dense, to the point of being almost unreadable. I would recommend reformatting this and, if space is not available, just state the main result of the proof and move the body to the supplementary materials. Also I am not sure that Proposition 1 is novel enough to warrant a proof here - presumably the theory of convergence for Sylvester equations is well-studied and the appropriate theorem could simply be cited here. I appreciate the inclusion of ablation studies. It should be more common in NIPS submissions. Overall, while I'm somewhat skeptical of how realistic the transductive ZSL setting is (since information about unseen classes is not likely to be available at training time) and I am not certain that the derivation of the updates is correct, the empirical performance does seem very good, and the results are a significant improvement on the aPY dataset.

Reviewer 4



Summary: In zero-shot learning for object recognition the system learns to recognize object types for which it saw no labeled images. It does this by learning projections between (or co-embedding) class label embeddings and image embeddings. This paper presents two relatively straightforward but experimentally effective innovations: (1) rather than learning a projection in one direction only (e.g. image embedding to class embedding), learn to project in both directions, somewhat like auto-encoders. (2) Induce a one-level class hierarchy by clustering the classes, and leverage the hierarchy as part of training. The authors also do some interesting optimization due to some transductive (min-min) optimization on the unlabeled data. The paper is clear and well written. There is extensive citation of past work (although it is possible that there are more relevant missing additional citations of which I unaware). The paper brings together some rather straightforward ingredients, in my opinion. Of course simultaneously learning projections in two directions, ala auto-encoders, is not new. It would be interesting to read commentary from the authors about other problem settings which have traditionally used one-direction-only projection learning but which the authors believe bi-directional would also be helpful. There are many, many examples of using a hierarchy of classes to improve classification. So again, this idea isn’t a big leap, but it is nice to see it working well again here. Simple isn’t a bad thing, especially when the method provides good experimental results, and this in my view is the most attractive strength of this paper. Across-the-board empirical improvements due to their method. So I’m most interested in understanding why it works better, and in “what data/problems it would be expected to help a lot or not at all”. I was happy to see the ablation studies. But I would have like to see much more error analysis, and analysis answering the question in quotes above. Could you construct a data set (via subset, or any other method) on which your method would not work well, and analyze/explain why it did not improve results? You tout your work on the min-min optimization. But I wondered why you learn with min-min rather than min-softmax, which might beneficially spread the gradient signal across the data. This sort of “soft assignment” is common in work on EM for semi-supervised learning in the 1990s by Kamal Nigam. You talk about using a hierarchy to improve classification, but your hierarchy seems pretty weak: just a flat clustering. Did you consider a hierarchical clustering? Even a simple method, such as hierarchical agglomerative clustering would produce a deep hierarchy. Small notes: Line 104: The first sentence in this line didn’t seem quite grammatical to me. Line 105: You are using citation references as grammatical entities in the sentence; better to use author names, and keep citation refs out of the English grammar. Line 147: I’m surprised that linear formulations outperform more complex non-linear ones. If you have space to say more here, that would be helpful.